# The Abdominal Pain Unit (APU). Study protocol of a standardized and structured care pathway for patients with atraumatic abdominal pain in the emergency department: A stepped wedged cluster randomized controlled trial

**Maria B. Altendorf**[1]*, **Martin Möckel**[2], **Liane Schenk**[1], **Antje Fischer-Rosinsky**[3], **Johann Frick**[1], **Lukas Helbig**[2], **Dirk Horenkamp-Sonntag**[4], **Dörte Huscher**[5], **Luisa Lichtenberg**[4], **Thomas Reinhold**[6], **Daniel Schindel**[1], **Britta Stier**[2], **Hanna Sydow**[6], **Yves-Noel Wu**[3], **Grit Zimmermann**[7], **Anna Slagman**[6]

1 Charité–Universitätsmedizin Berlin, Institute of Medical Sociology and Rehabilitation Science, Berlin, Germany, 2 Emergency and Acute Medicine (CVK, CCM), Charité—Universitätsmedizin Berlin, Berlin, Germany, 3 Health Services Research in Emergency Medicine; Emergency and Acute Medicine (CVK, CCM), Charité—Universitätsmedizin Berlin, Berlin, Germany, 4 Techniker Krankenkasse, Hamburg, Germany, 5 Charité–Universitätsmedizin Berlin, Institute of Medical Biometry and Clinical Epidemiology, and Berlin Institute of Health, Berlin, Germany, 6 Charité–Universitätsmedizin Berlin, Institute of Social Medicine, Epidemiology and Health Economics, Berlin, Germany, 7 TMF—Technology, Methods, and Infrastructure for Networked Medical Research, Berlin, Germany

* maria.altendorf@charite.de

## Abstract

This study aims to improve emergency department (ED) care for patients suffering from atraumatic abdominal pain. An application-supported pathway for the ED will be implemented, which supports quick, evidence-based, and standardized diagnosis and treatment steps for patients with atraumatic abdominal pain at the ED. A mixed-methods multicentre cluster randomized controlled stepped wedge trial design will be applied. A total of 10 hospitals with EDs (expected n = 2.000 atraumatic abdominal pain patients) will consecutively (every 4 months) be randomized to apply the intervention. Inclusion criteria for patients are a minimum age of 18 years, suffering from atraumatic abdominal pain and being insured with a German statutory health insurance. Primary outcomes: acute pain score at time of discharge from ED, duration of treatment at the ED, patient-reported satisfaction. Secondary endpoints include patient safety and quality of care parameters, process evaluation parameters, and costs and cost-effectiveness parameters. Quantitative data will be gathered from patient-surveys, clinical records, and routine data from hospital information systems as well as from a participating German statutory health insurance. Descriptive and analytic statistical analysis will be performed to provide summaries and associations for primary patient-reported outcomes, process measures, quality measures, and costs. Qualitative data collection consists of participatory patient observations and semi-structured expert interviews, which will be inductively analysed. Findings will be disseminated in publications

**Data Availability Statement:** No datasets were generated or analysed during the current study. All relevant data from this study will be made available upon study completion.

**Funding:** MM received funding from the Innovation Funds (https://innovationsfonds.g-ba.de/) from the German Federal Joint Committee (G-BA) under the grant number 01NVF19025 for this project. The funders had and will not have a role in study design, data collection and analysis, decision to publish, or preparation of the manuscript.

**Competing interests:** The authors have declared that no competing interests exist.

**Abbreviations:** APU, Abdominal Pain Unit; cRCT, Cluster randomized controlled trial; ED, Emergency department; ICU, Intensive care unit; NRS, Numeric rating scale; qSOFA, quick Sequential (Sepsis-Related) Organ Failure Assessment; SHI, Statutory health insurance; TK, Techniker Krankenkasse (the participating German statutory health insurance); ZUF-8, Züricher Patientenzufriedenheit Fragebogen.

in peer-reviewed journals, on conferences, as well as via a project website. To ensure data protection, appropriate technical and organisational measures will be taken.

**Trial registration:** DRKS00021052.

## Introduction

A common complaint in patients presenting in the emergency department (ED) is atraumatic abdominal pain with a prevalence of 5–20% per year [1–3]. Patients with atraumatic abdominal pain can have a very broad range of diagnoses, ranging from flatulencies to more serious underlying diseases, such as an acute pancreatitis. Most probably, due to the difficulty to quickly diagnose and treat patients with acute abdominal pain, the associated hospital mortality rate of 5.1% is relatively high, e.g. compared to chest pain (with a 0.9% hospital mortality) [1]. Among patients aged 65 years or older mortality even increases [4]. Thus, as Berner and Dormann stated [5] every patient presenting with atraumatic abdominal pain in the ED should be seen as a high-risk patient who is in urgent need for fast diagnosis and treatment. To date, no standardized care pathway for patients with atraumatic abdominal pain exists, why finding a diagnosis for those patients depends on each hospital ward's internal standard of care, as well as each physician's qualifications and experience [5]. To assure high quality of care and potentially reduce mortality, it seems imperative to implement a novel management pathway, which standardizes the process from start of care to final diagnosis, disposition and specific therapy of atraumatic abdominal pain patients in the ED. Therefore, a team of multidisciplinary experts has developed the „Abdominal Pain Unit"(APU) treatment process based on the Delphi method. The APU-process will be digitally supported by an application software (i.e. the APU-App). The App-supported APU-process aims to improve patients' care by:

1. Leading to a shorter duration of treatment in the ED while improving patient-reported outcomes (assessed as acute pain score or/and patient satisfaction) at discharge from the ED; or

2. Improving patient-reported outcomes (assessed as acute pain score or/and patient satisfaction) at discharge from the ED while measuring a constant duration of treatment in the ED; or

3. Leading to a shorter duration of treatment in the ED and unchanged patient-reported outcomes (assessed as acute pain score and patient satisfaction) at discharge from the ED.

## Materials and methods

### Study design

The study design and a schedule of activities prior to trial and during the trial is illustrated in Fig 1.

A mixed-methods, multicentre cluster-randomized controlled stepped wedge trial (cRCT) will be applied (see Fig 2). Patients in the control group will receive care as usual, which will be according to the standard of care of each participating hospital and might differ between study sites. Patients enrolled in the intervention group will receive treatment according to the APU-process. Since the implementation of the APU-process will affect the entire ED, individual randomization of patients within centres is not feasible. Therefore, an alternative approach was chosen: the introduction of the process in the 10 study centres will take place in clusters of

| | Enrolment | Allocation | STUDY PERIOD | | | | Close-out |
| --- | --- | --- | --- | --- | --- | --- | --- |
| | | | Post-allocation | | | | |
| TIMEPOINT | $t_0$ | n.a.* | 12 months preceding study | $t_0$ | $t_1$ | During course of study (12 months) | After last patient in |
| **ENROLMENT:** | | | | | | | |
| Eligibility screen | X | | | | | | |
| Informed consent | X | | | | | | |
| Allocation | | X | | | | | |
| **INTERVENTIONS:** | | | | | | | |
| *Control* | X | X | | | | | |
| *APU Intervention* | X | X | | | | | |
| **ASSESSMENTS**\*\***:** | | | | | | | |
| *Acute pain score* | | | | X | | | |
| *Duration of treatment at the ED* | | | | X | X | | |
| *Patient satisfaction* | | | | X | X | | |
| *Process quality* | | | | X | X | | |
| *Qualitative data* | | | | | | X | |
| *Costs & Cost-effectiveness* | | | X | X | X | X | |

**Fig 1. Schedule of enrolment, intervention, and assessments.**

centres. As illustrated in Fig 2, this stepped wedge design [6–8] involves a sequential, randomized, consecutive transfer of clusters (every four months) from the control arm to the intervention arm until finally all clusters have implemented the new process. In this way, equal periods

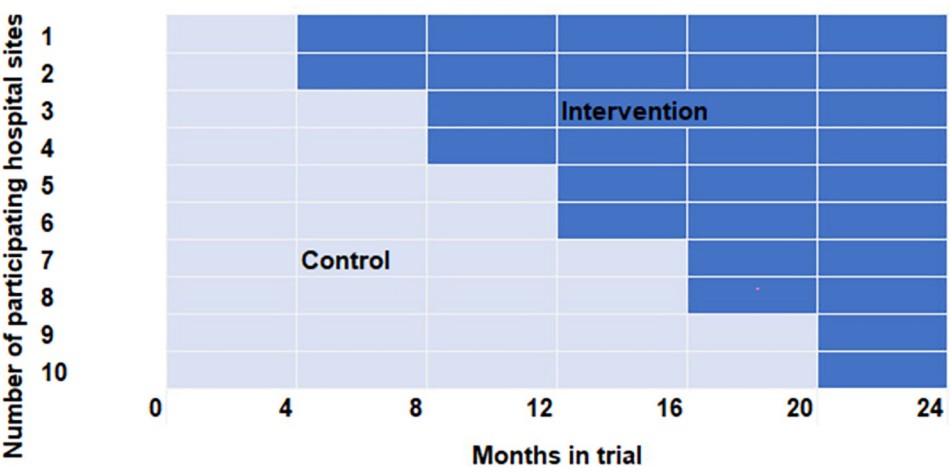

**Fig 2. Stepped wedge trial design.**

of control and intervention can be observed concerning all clusters but length of control and interventions period differs between different clusters.

Ethical approval was obtained from Charité's responsible institutional review boards (EA2/219/20). Prior to study participation, written informed consent will be obtained from all study participants. Findings will be disseminated in publications in peer-reviewed journals, on conferences, as well as via a project website. To ensure data protection, appropriate technical and organisational measures will be taken.

Data collection is set to a time period of two years, until all clusters (n = 10) will have implemented the intervention for at least four months. Medical staff in each ED will receive thorough training of the APU-process immediately prior to intervention implementation in the ED.

## The intervention

The App-supported APU-process (in the following and for clarity, the term APU-process will be used) starts with a patient who suffers from atraumatic abdominal pain presenting in the ED. By means of the APU-process, physicians will be supported in making a structured decision for the subsequent diagnostic and treatment process. Eventually, the APU-process ends with either the discharge of the patient with a sufficiently accurate diagnosis from the ED or the patient being admitted to another hospital unit for further treatment. Patients with a shock syndrome or sepsis leave the path for special intensive care early. The first step in the APU-process includes a medical history, a medical examination, measurement of blood parameters, and pain management for patients with atraumatic abdominal pain. In a second step, a re-evaluation will lead to a decision whether the patient will be discharged from the ED to ambulant care (i.e. in case of unsuspicious clinical findings) or if further diagnostic measures have to be taken. Thus, in a third step, the patient will receive a sonography, however, in the case of persistently unclear clinical findings, in a fourth step, either additional imaging methods will be used, such as computer tomography or magnetic resonance imaging, or a multi-disciplinary consultation and if necessary a patient observation for a few hours will be performed. Patients with worsening medical condition will leave the path for intensive care.

## Measures

Quantitative and qualitative data will be collected to facilitate cross verification of data and to enhance credibility of the results [9]. Primary and secondary outcomes are presented in Table 1.

The primary outcome, duration of treatment in the ED, is measured as the timespan between the time point when the patient registers at the ED desk with his/her health insurance card and the documented time point when the physician decides that the patient can leave the ED (discharge, t0). Once the patient is enrolled to the APU-study, secondary outcomes, such as patient safety and quality of care parameters (e.g. quality of life) and process parameters (e.g. time until diagnostic measures are available) will be collected. At discharge from the ED (t0) the study nurse will assess the remaining primary (patient-reported) outcomes, acute pain score and patient satisfaction, as well as remaining patient-reported secondary outcomes (e.g. demographics, subjective health status). At 30-days follow-up (t1), study nurses will follow-up on patient-reported outcomes (see Table 1) via phone or will send participants the URL to access an online survey.

Besides a quantitative effectiveness and process evaluation of the APU-process from patient questionnaires, hospital patient-records regarding acute pain scores, duration of treatment in the ED, and patient satisfaction among the targeted 2.000 patients, also a qualitative process

**Table 1. Overview of primary & secondary outcomes.**

| Outcome | Concept | Instrument / parameter | Time point of measuring (Module) |
|---|---|---|---|
| *Primary outcomes* | | | |
| | Acute pain score | NRS [10] [a] | t0 (Module 1) |
| | Duration of treatment at ED | Timespan between beginning and end of treatment in the ED | t0 (Module 2) |
| | Patient satisfaction | Züricher Patientenzufriedenheit Fragebogen (ZUF-8; [11]) [a] | t0 (Module 1) |
| *Secondary outcomes* | | | |
| | Quality of care / patient safety indicators | E.g. Quality of life (EUROHIS-QOL-8 [12, 13]); Mortality, course of treatment in the hospital (e.g. inpatient stay following the ED treatment, ICU stay, duration of ICU stay) | t0 & t1 (Module 1) t0 & t1 (Module 2, 3) |
| | Process quality | Process times (e.g. time until diagnostic measures are available / diagnostic examination, frequency of diagnostic procedures); Qualitative data from semi-structured expert interviews & participatory observations: Potential success factors & pitfalls of the APU-process, feasibility in routine care, applicability in routine care | t0 (Module 2) During course of study (Module 5) |
| | Costs and Cost-effectiveness | Costs of: hospital stays, outpatient visits, medication/pharmaceuticals, adjuvants and devices, total costs in relation to primary outcomes | 12 months preceding the ED stay & during course of study (Module 2, 3, 4) |

Note. ED = Emergency Department. NRS = Numeric rating scale; subjective measure for rating the pain on an eleven-point numeric scale from 0 (no pain at all) to 10 (worst imaginable pain). ZUF-8 = Züricher Patientenzufriedenheit Fragebogen. EUROHIS-QOL-8 = measure for Quality of Life, derived from the WHOQOL-100 and the WHOQOL-BREF; for this study, one item was extracted from the full scale.

[a] = assessed by study nurse. ICU = Intensive care unit.

analysis will be performed. The qualitative process analysis consists of participating observations (n = 25) and expert interviews (n = 35), in order to gain an understanding of the different perspectives and contexts, as well as the facilitators and barriers of the implementation of the APU-process. In addition, a cost and cost-effectiveness analysis will be performed based on health insurance data from the participating SHI.

## Inclusion and exclusion criteria

Patients being 18 years or older presenting at the participating EDs with atraumatic abdominal pain will be screened for eligibility in the study. If patients have a legal representative, this representative has to agree to participation in the study and patients need to be insured with a German SHI for inclusion in the present study. Exclusion criteria are traumatic causes of abdominal pain, suspicion of sepsis (quick Sequential (Sepsis-Related) Organ Failure Assessment: qSOFA score $\geq$ 2 [14]) or suspicion of shock (shock index $\geq$ 1), and insufficient knowledge of German language.

## Sampling, sample size and power calculation

Physicians working in the ED will recruit eligible patients. Specially trained study nurses will support the recruitment of patients and the data collection in the EDs (e.g., administering patient surveys, extracting data from hospital records). For qualitative data analysis, experts will be recruited at staff-trainings for the APU-process, as well as via email invitations.

As illustrated in Fig 3, for the trial period of two years, the average number of patients presenting with atraumatic abdominal pain in the ED in German hospital sites is expected to be between 1.750–3.500 patients [1].

An a-priori power calculation based on the primary endpoints (acute pain score at discharge from ED, duration of treatment at the ED, patient satisfaction) with nQuery Advisor 7.0 [15] and corrected for the stepped wedge-design with 10 centers confirmed that a total

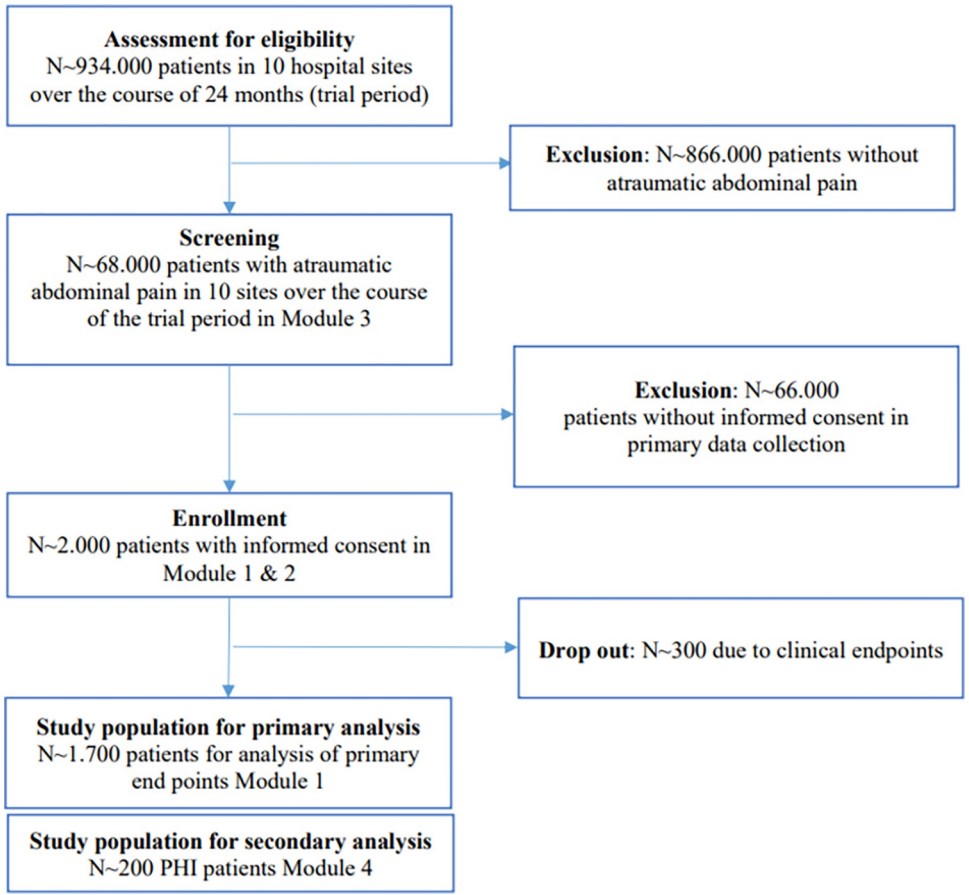

**Fig 3. Flow chart of patients admitted to the participating 10 emergency departments in Germany and resulting study population with atraumatic abdominal pain.**

sample size of n = 1.700 patients (which equals about n = 200 patients per study arm) is sufficient for analyzing the three endpoints in parallel adjusted for multiple testing; considering the potentially 15% study attrition rate the enrollment should aim for 2.000 patients.

## Evaluation modules & data management plan

Recruitment of participants takes place in the EDs of all 10 involved German hospital sites by medical physicians. Study nurses will support baseline data collection (t0) in the EDs (i.e. administering patient surveys and documentation), and will carry out the 30-days follow-up data collection (t1). The evaluation process, which is divided in five evaluation modules, is illustrated in Fig 4.

**Module 1: Patient-reported outcomes.** Patients who present in the ED with atraumatic abdominal pain will be identified by the physician and asked by physicians to participate in the APU study. At discharge or transfer from the ED (t0), study nurses will collect patient-reported outcomes, i.e. acute pain scores (numerical rating scale, NRS 0–1 [10]), patient satisfaction (ZUF-8, [11, 16]), quality of life (EUROHIS-QoL-8, [12, 13]), and socio-economic data with tablets (i.e. digitally and online) or with a paper-pencil-manner. At 30 days post-ED admission (t1), patients will be followed-up on their acute pain score, their quality of life, and other care-related outcomes. Either the study nurses will send patients a link to access the

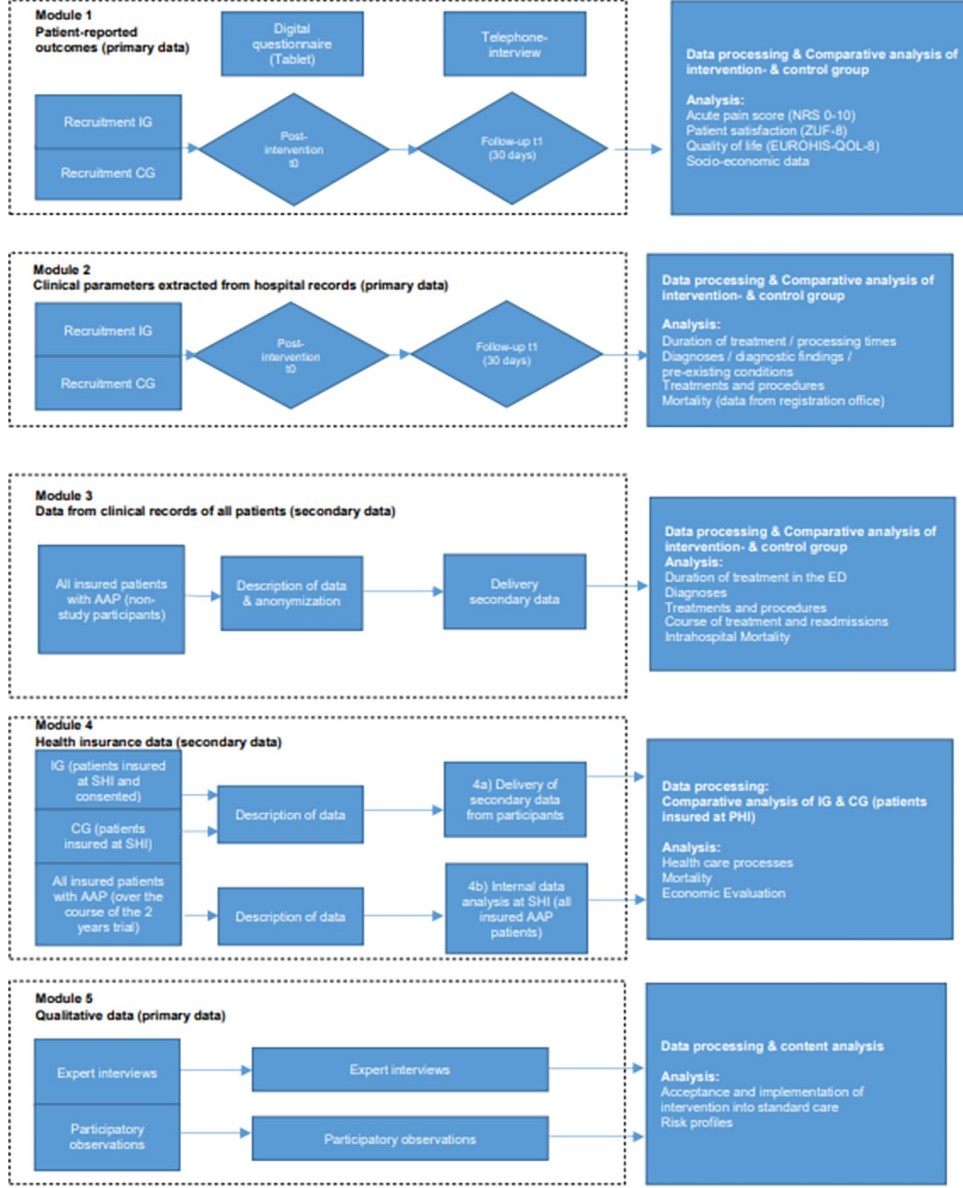

**Fig 4. Evaluation modules.**

follow-up survey online, or patients will be followed-up via telephone, or perform an in-hospital follow-up in case a patients is still hospitalized.

**Module 2: Primary health care data.** Study nurses will extract consented patient's clinical parameters (e.g. vital parameters, pre-existing conditions, onset of pain, and procedures and results) and duration of treatment in the ED from hospital records to an electronic Case Report Form at t0 and t1. Moreover, at t1, re-admission data will be extracted.

**Module 3: Secondary data from clinics.** Routinely collected data from all patient suffering from atraumatic abdominal pain (identified by physicians), such as transport, timestamps, diagnosis, vital parameters, and blood parameters, as well as data from the potentially subsequent inpatient stay of all patients with atraumatic abdominal pain treated in the ED during the two-year study period will be extracted. Those data will be de facto anonymized.

**Module 4: Secondary health insurance data.** Module 4 consists of two different parts of data extraction, namely part a) and b):

a. Patients who consented to participate in the APU study and are also insured with the participating health insurance company, will be asked to agree to the provision of routinely collected data by the insurance company to the evaluating institute. These data include frequency and amount of health care resource consumption, health data, such as diagnosis and associated health care costs one year prior to trial until t1. These data will be economically investigated by the evaluating institute from the perspective of the health insurance company in terms of costs and cost-effectiveness (total costs related to primary outcomes) of the APU-process compared to controls over the whole study duration. In addition, an economic evaluation from the perspective of the hospital will be performed to investigate whether the usage of resources in the ED might have changed. For this analysis, data collected in Module 3 will be used (e.g., the number of ED procedures).

b. Moreover, the participating health insurance company will perform an internal evaluation of effects from data of all insured patients with atraumatic abdominal pain syndrome at the same duration (i.e. one year prior to trial until t1).

**Module 5: Qualitative process evaluation data.** Experts (n = 35, i.e. physicians working with the APU-process in the EDs) and patients from the intervention group (n = 25) enrolled in the APU study will be identified through purposive sampling. Interviews will be conducted at all locations. Prior to expert interviews, participatory patient observations will be conducted and documented as field notes which will subsequently be transferred into standardized observational protocols. Semi-structured interviews will be conducted via phone or face-to-face. Preceding the interviews, written informed consent will be given by participants.

## Data analysis plan

Descriptive and associated statistical analysis will be performed to provide summaries and analytical results for patient-reported outcomes, process measures, quality measures and costs. The three primary outcomes will be analyzed as planned for the RCT implementing adjustment methods necessary for data received in a stepped wedge study design. Generalized linear mixed models (GLMM) will be used to analyze primary outcomes as these allow also non-normal standard deviations and binary outcomes. Moreover, GLMM can compensate for differing cluster size (e.g. number of participants differs between study sites). The detailed SAP is currently prepared based on information about the data structure provided by the clinical team and the data managing team. In case of unequally distributed confounders (e.g., age, gender case mix) or potential risk factors (e.g., smoking or post-operative state of patient) in the data, sensitivity analysis will be performed.

Primary qualitative data collected in Module 5, consisting of field notes from participatory observations and the transcripts from semi-structured expert interviews will be analyzed inductively.

## Dissemination

The project will be implemented under the consortium leadership of the Emergency and Acute Medicine, Campus Mitte und Virchow Klinikum Charité. The scientific evaluation is led by the Institute of Medical Sociology and Rehabilitation Science and the Institute of Health Services Research in Emergency Medicine together with the Institute of Medical Biometrics and Clinical Epidemiology and the Institute of Social Medicine, Epidemiology and Health Economics of the Charité–Universitätsmedizin Berlin. TMF e.V. is responsible for data protection

aspects, such as the creation and coordination of the data protection concept, as well as for aspects regarding medical device regulations.

## Discussion of potential strengths and limitations

Despite the potential limitation that a comprehensive implementation of the App-supported APU-process in all German hospitals will be challenging, since technological barriers could potentially arise, such as depending on the Wi-Fi-connection, slow response time, or system down time, a universally deployable application will be developed to efficiently guide ED care of atraumatic abdominal pain in the future. It is assumed that findings of this study are generalizable and can be broadly transferred into practice since hospitals included in this study are of different size, and location in Germany.

However, based on the exclusion criteria that participants need to have sufficient German language proficiency, as well as have to consent to participation either themselves or by their legal representative, migrant population or elderly with advanced dementia might be underrepresented in this study. For instance, especially for the migrant population, the experience of pain differs culturally [17]. Potentially excluding these patient groups from this study might subsequently lead to an underrepresentation of these vulnerable groups.

However, to generate highest possible evidence in the APU-study, sophisticated data triangulation from different sources will be used. In this triangulation process data will be linked from individual, patient-reported outcomes with secondary, objective data from clinical routine documentation and health insurers as well as qualitatively gathered data from participatory observations and expert interviews.

Another point for discussion is the potential risk of an ,over-diagnosis', due to physicians blindly following the App-supported APU-process. In that case, patients could have a higher exposure of imagining diagnostics and subsequently a higher exposure to radiation or a longer duration of stay in the ED. This could also lead to a potential increase of inpatient stays as well as higher use of resources. Yet, no German standard of care for patients with atraumatic abdominal pain in the ED exists, why such over-diagnosis are potentially currently happening as well. The novel APU-process aims to offer clear indications for when, for instance, imaging diagnostics need to be taken, which on the contrary could prevent over-diagnosis. Further, it needs to be mentioned that based on the current absence of common German care standards for patients with atraumatic abdominal pain in the ED, major differences between the control groups of the participating centres might become a potential challenge for data analysis in this study. The control group in this study consists of care as usual, which is a different standard of care, diagnosis and treatment of patients with atraumatic abdominal pain in each study site, depending on the physician working in the ED. Another reason, why implementation of an evidence-based care standard, namely the APU, seems imperative for the best possible quality of care for this group of patients.

It is expected that the APU-process will have high potential to improve quality of acute care and quality of life for patients with atraumatic abdominal pain by a standardized diagnosis and treatment pathway. Moreover, we expect that patients´ experience of treatment in the ED will improve as a consequence of potentially shorter treatment times and faster diagnosis. As a result of this study, the standardized APU-process is expected to be ready to be scaled up nationwide.

## Supporting information

**S1 Checklist. SPIRIT 2013 Checklist: Recommended items to address in a clinical trial protocol and related documents**∗**.**
(DOCX)

**S1 Protocol. Antrag auf Beratung durch die Ethikkommission zur Durchführung eines medizinisch-wissenschaftlichen Vorhabens, welches weder die klinische Prüfung eines Arzneimittels noch Medizinproduktes beinhaltet.**
(PDF)

**S2 Protocol. Application for advice by the ethics committee on the implementation of a medical-scientific project that does not involve the clinical testing of a medicinal product or medical device.**
(DOCX)

## Author Contributions

**Conceptualization:** Martin Möckel, Liane Schenk, Lukas Helbig, Dörte Huscher, Thomas Reinhold, Daniel Schindel, Anna Slagman.

**Formal analysis:** Dörte Huscher.

**Funding acquisition:** Martin Möckel, Liane Schenk, Anna Slagman.

**Methodology:** Liane Schenk, Antje Fischer-Rosinsky, Johann Frick, Dörte Huscher, Thomas Reinhold, Daniel Schindel.

**Project administration:** Liane Schenk, Antje Fischer-Rosinsky, Britta Stier, Anna Slagman.

**Resources:** Martin Möckel, Liane Schenk, Anna Slagman.

**Supervision:** Martin Möckel, Liane Schenk, Luisa Lichtenberg, Grit Zimmermann, Anna Slagman.

**Writing – original draft:** Maria B. Altendorf.

**Writing – review & editing:** Maria B. Altendorf, Martin Möckel, Liane Schenk, Antje Fischer-Rosinsky, Johann Frick, Lukas Helbig, Dirk Horenkamp-Sonntag, Dörte Huscher, Luisa Lichtenberg, Thomas Reinhold, Daniel Schindel, Britta Stier, Hanna Sydow, Yves-Noel Wu, Grit Zimmermann, Anna Slagman.

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
