## [Decision Letter · Decision Letter 0]

3 May 2022

PONE-D-21-38548The Abdominal Pain Unit (APU). Study protocol of a standardized and structured care pathway for patients with atraumatic abdominal pain in the emergency department: A stepped wedged cluster randomized controlled trial.PLOS ONE

Dear Dr. Altendorf,

Thank you for submitting your manuscript to PLOS ONE. After careful consideration, we feel that it has merit but does not fully meet PLOS ONE’s publication criteria as it currently stands. Therefore, we invite you to submit a revised version of the manuscript that addresses the points raised during the review process.

We look forward to receiving your revised manuscript.

Kind regards,

Steven Eric Wolf, MD

Academic Editor

PLOS ONE

“This work was supported by the Innovation Funds from the German Federal Joint Committee (G-BA) under the grant number 01NVF19025.”

“MM received funding from the Innovation Funds (https://innovationsfonds.g-ba.de/ ) from the German Federal Joint Committee (G-BA) under the grant number 01NVF19025 for this project. The funders had and will not have a role in study design, data collection and analysis, decision to publish, or preparation of the manuscript.”

Additional Editor Comments:

Editor - Thank you for submitting your paper to us for review. I sent it to seven distinguished referees for comment and decision of whom two agreed to review; you will see these below. They thought that the paper has merit, but each have raised some substantial issues to be addressed in a revision. Please carefully consider the comments below and reply directly to each in a cover letter with appropriate marked and linked changes to the manuscript. I look forward to seeing the revision, which I will send back to the same referees for further comment and decision. Please understand that this is not a guarantee of future publication, as the revision must stand on its own merit.

Reviewers' comments:

Reviewer's Responses to Questions

**Comments to the Author**

1. Does the manuscript provide a valid rationale for the proposed study, with clearly identified and justified research questions?

Reviewer #1: Yes

Reviewer #2: Yes

2. Is the protocol technically sound and planned in a manner that will lead to a meaningful outcome and allow testing the stated hypotheses?

Reviewer #1: Yes

Reviewer #2: Partly

3. Is the methodology feasible and described in sufficient detail to allow the work to be replicable?

Reviewer #1: Yes

Reviewer #2: Yes

4. Have the authors described where all data underlying the findings will be made available when the study is complete?

Reviewer #1: Yes

Reviewer #2: Yes

5. Is the manuscript presented in an intelligible fashion and written in standard English?

Reviewer #1: Yes

Reviewer #2: Yes

6. Review Comments to the Author

You may also provide optional suggestions and comments to authors that they might find helpful in planning their study.

Reviewer #1: The paper references a possibility of ‘over-diagnosis’ and over-exposure to radiation. When subjective information is protocolized, there is lots of variation and misdiagnosis.

Guidelines are molded in the vein of being a clinician and coming to a diagnosis. The general pathway of medicine is history and physical exam then laboratory data/imaging to confirm the diagnosis. A guideline then provides treatment modalities based on the diagnosis and patient condition.

The fear of creating a standardized protocol for the emergency department may lead to less clinical judgement and more reliance on following a standardized practice that does not individualize the patient.

It would be good to see a protocol that could help triage the severity of the diagnosis and how not to delay consultant involvement. Severe pancreatitis was referenced in the paper, and it can be a very deadly disease. To diagnosis the diseases you need two of the following 1) Typical pain presentation 2) Elevated lipase greater than 3 times the upper limit of normal 3) imaging (CT or US) finding of pancreatic inflammation. Epigastric pain that radiates to the back may present in these patients, but is also found in ruptured aortic aneurysms (which is acutely more lethal). Would this protocol recognize this difference in severity? Early recognition of a diagnosis and resuscitation is what reduces mortality. This is done initially with a thorough history and physical. Imaging and labs only confirm diagnosis and should not be the leading factor in initial management of critical patients.

Reviewer #2: the authors have embarked on a worthwhile albeit difficult journey. i am concerned that time in the ED is the primary endpoint. Time in the ed is dependent on so many variables, often not related directly to improved patient care or outcome. i would suggest the authors focus on a more patient centered outcome that is indicative of improved care, rather than a process measure. additionally there is little mention of how pain will be measured and controlled. Please describe

7. PLOS authors have the option to publish the peer review history of their article (what does this mean?). If published, this will include your full peer review and any attached files.

Reviewer #1: No

Reviewer #2: No

---

## [Author Response · Author response to Decision Letter 0]

18 May 2022

Response to review on manuscript “The Abdominal Pain Unit (APU). Study protocol of a standardized and structured care pathway for patients with atraumatic abdominal pain in the emergency department: A stepped wedged cluster randomized controlled trial.” (PONE-D-21-28548)

We would like to thank the reviewer for his/her thorough review of our manuscript and the constructive, relevant remarks and suggestions. In the following, we will try to explain how we solved remarks indicated by the reviewer. The remarks of the reviewer are depicted first, followed by our responses in italics.

Reviewer's comments:

Reviewer #1:

The fear of creating a standardized protocol for the emergency department may lead to less clinical judgement and more reliance on following a standardized practice that does not individualize the patient. It would be good to see a protocol that could help triage the severity of the diagnosis and how not to delay consultant involvement. 

 We thank the reviewer for his/her critical thoughts on the tested standardized APU-process and we agree that a standardized treatment process may have its benefits and limitations. The reviewer fears that a standardized treatment process, such as the APU-process could potentially lead to less clinical judgement and more reliance on the process. We argue that treatment guidelines or standardized processes do not lead to potential delay in consultation and/or treatment. Other standardized treatment processes, such as the chest pain unit (1, 2), show very promising indications that such processes can improve processes and patient-centered outcomes are also standardized processes within the clinical setting. Moreover, physicians follow clinical guidelines for certain diagnoses, which differ between hospital(s wards). Also, as mentioned in the manuscript on page 4 & 5, line 89 ff.: “[…] no standardized care pathway for patients with atraumatic abdominal pain exists, why finding a diagnosis for those patients depends on […] each physician’s qualifications and experience (3).” As described in our manuscript (page 5, line 96 f.), the APU-process was developed by a multidisciplinary team of experts in a Delphi process (4) and therefore, seems very promising in increasing patient safety, patient satisfaction with the care provided and subsequently reducing mortality.

It is important to note that there are no mandatory automatisms in the APU treatment process. The goal of the APU-process is to give guidance to physicians in a very complex field of emergency medicine. All final decisions in the treatment of atraumatic abdominal pain – even within the APU-process - are made by the physician. The APU-app guides-process can be seen as a checklist, which only supports physicians to not miss crucial aspects in treatment and diagnosis while being focused on clinical judgement. 

Severe pancreatitis was referenced in the paper, and it can be a very deadly disease. To diagnosis the diseases you need two of the following 1) Typical pain presentation 2) Elevated lipase greater than 3 times the upper limit of normal 3) imaging (CT or US) finding of pancreatic inflammation. Epigastric pain that radiates to the back may present in these patients, but is also found in ruptured aortic aneurysms (which is acutely more lethal). Would this protocol recognize this difference in severity? Early recognition of a diagnosis and resuscitation is what reduces mortality. This is done initially with a thorough history and physical. Imaging and labs only confirm diagnosis and should not be the leading factor in initial management of critical patients.

We thank the reviewer for the attentive question whether the APU treatment process, may recognize the difference in severity, as for instance, between a ruptured aortic aneurism and a pancreatic inflammation. The APU treatment process is specifically designed to not miss crucial, time-critical diagnoses. This is ensured by various checks within the treatment process. The first step in the APU-process ensures that the attending physician recognizes signs of clinical instability, shock or sepsis based on vital parameters (like blood pressure, vigilance, heart rate) and clinical appearance. At the beginning of the APU-process, every patient receives an ECG to warrant that no ST-Elevation myocardial infarction is overlooked, which may present as upper abdominal pain. In a next and second step of the APU-process, a thorough history and physical examination, as well as pain management and abdominal lab will be executed. Then, as a result of the second step, a clinical evaluation of the patient’s condition will be done and again vital parameters will be checked. These steps have been designed to ensure that crucial diagnoses will be recognized as early as possible. As essential part of the APU-process, at many time points, physicians are reminded to watch out for “red flags” to guarantee, that they do not miss time-critical diagnoses, such as ruptured abdominal aortic aneurysm, incarcerated hernia, testicular torsion, hollow organ perforation, ileus, mesenteric ischemia, splenic rupture and myocardial infarction. 

Opposed to the reviewer’s remark that imaging and labs might be leading factors in the initial management of critical patients, in the APU-process these two tools are only used to confirm or reject certain diagnoses.

Reviewer #2: 

The authors have embarked on a worthwhile albeit difficult journey. i am concerned that time in the ED is the primary endpoint. Time in the ed is dependent on so many variables, often not related directly to improved patient care or outcome. i would suggest the authors focus on a more patient centred outcome that is indicative of improved care, rather than a process measure. 

 We thank the reviewer for this careful observation about potential concerns regarding the measurement and interpretation of the parameter time in the ED (i.e. we defined this parameter as “duration of treatment in the ED”). We agree with the reviewer that the parameter “duration of treatment in the ED” depends on various parameters, such as the triage level, chief complaint, and mode of arrival (5), and might therefore only partially illustrate improved care. However, a shortened duration of treatment can directly benefit patients regarding shorter waiting times for diagnostics and treatments. This is, because medical personal might be readily available and not engaged with other patients.

However, as suggested by the reviewer, we also do focus on patient-centered outcomes (see page 5, line 98 ff.), to have a primary, aggregated aim of our study (consisting of three hypotheses): 

“1) Leading to a shorter duration of treatment in the ED while improving patient-reported outcomes (assessed as acute pain score or/and patient satisfaction) at discharge from the ED; or

2) Improving patient-reported outcomes (assessed as acute pain score or/and patient satisfaction) at discharge from the ED while measuring a constant duration of treatment in the ED; or

3) Leading to a shorter duration of treatment in the ED and unchanged patient-reported outcomes (assessed as acute pain score and patient satisfaction) at discharge from the ED.”

Thus, we can highlight that we also measure patient-centered outcomes as main/primary endpoints. Namely: Patient-reported outcomes: patient satisfaction and acute pain score. 

Moreover, it is to be mentioned that by combining three different outcomes, it is possible that lower assesses pain score and higher patient satisfaction alone can be interpreted as an improvement of patient’s care, while the duration of treatment in the ED stays constant.

Additionally there is little mention of how pain will be measured and controlled. Please describe 

With regards to the reviewer’s remark on the measurement of pain, we have now added a description of the measurement of pain with the NRS below Table 1 (page 8, line165 ff.): “NRS = Numeric rating scale; subjective measure for rating the pain on an eleven-point numeric scale from 0 (no pain at all) to 10 (worst imaginable pain).” 

References

1. Breuckmann F, Rassaf T, Hochadel M, Giannitsis E, Munzel T, Senges J. German chest pain unit registry: data review after the first decade of certification. Herz. 2021;46(Suppl 1):24-32.

2. Keller T, Post F, Tzikas S, Schneider A, Arnolds S, Scheiba O, et al. Improved outcome in acute coronary syndrome by establishing a chest pain unit. Clin Res Cardiol. 2010;99(3):149-55.

3. Berner L, Dormann H. Unclear abdominal pain in central emergency admissions. An algorithm. Med Klin Intensivmed Notfmed. 2013;108(1):33-40.

4. Helbig L, Stier B, Romer C, Kilian M, Slagman A, Behrens A, et al. [The abdominal pain unit as a treatment pathway : Structured care of patients with atraumatic abdominal pain in the emergency department]. Med Klin Intensivmed Notfmed. 2021.

5. Ding R, McCarthy ML, Desmond JS, Lee JS, Aronsky D, Zeger SL. Characterizing waiting room time, treatment time, and boarding time in the emergency department using quantile regression. Acad Emerg Med. 2010;17(8):813-23.

---

## [Decision Letter · Decision Letter 1]

1 Jul 2022

PONE-D-21-38548R1The Abdominal Pain Unit (APU). Study protocol of a standardized and structured care pathway for patients with atraumatic abdominal pain in the emergency department: A stepped wedged cluster randomized controlled trial.PLOS ONE

Dear Dr. Altendorf,

Thank you for submitting your manuscript to PLOS ONE. After careful consideration, we feel that it has merit but does not fully meet PLOS ONE’s publication criteria as it currently stands. Therefore, we invite you to submit a revised version of the manuscript that addresses the points raised during the review process.

We look forward to receiving your revised manuscript.

Kind regards,

Steven Eric Wolf, MD

Academic Editor

PLOS ONE

Journal Requirements:

Reviewers' comments:

Reviewer's Responses to Questions

**Comments to the Author**

1. Does the manuscript provide a valid rationale for the proposed study, with clearly identified and justified research questions?

Reviewer #1: Yes

Reviewer #2: Yes

2. Is the protocol technically sound and planned in a manner that will lead to a meaningful outcome and allow testing the stated hypotheses?

Reviewer #1: Yes

Reviewer #2: Yes

3. Is the methodology feasible and described in sufficient detail to allow the work to be replicable?

Reviewer #1: Yes

Reviewer #2: Yes

4. Have the authors described where all data underlying the findings will be made available when the study is complete?

Reviewer #1: Yes

Reviewer #2: Yes

5. Is the manuscript presented in an intelligible fashion and written in standard English?

Reviewer #1: Yes

Reviewer #2: Yes

6. Review Comments to the Author

You may also provide optional suggestions and comments to authors that they might find helpful in planning their study.

Reviewer #1: No further suggestions. Having a this protocol can be very helpful in the emergency setting.

Reviewer #2: A minor comment,... The authors focussed on pain control, patient satisfaction and time in the ED. However, how will this insure that improving these outcomes also will improve standard quality of care outcomes? Please address this in the paper.

7. PLOS authors have the option to publish the peer review history of their article (what does this mean?). If published, this will include your full peer review and any attached files.

Reviewer #1: No

Reviewer #2: No

---

## [Author Response · Author response to Decision Letter 1]

13 Jul 2022

We would like to thank the reviewer for his/her thorough review of our manuscript and the constructive, relevant remarks and suggestions. In the following, we will try to explain how we solved remarks indicated by the reviewer. The remarks of the reviewer are depicted first, followed by our responses in italics.

Reviewer's comments:

Reviewer #2:

A minor comment,... The authors focussed on pain control, patient satisfaction and time in the ED. However, how will this insure that improving these outcomes also will improve standard quality of care outcomes? Please address this in the paper.

 We thank the reviewer for his/her critical thoughts on the outcomes of the APU study and the improvement of quality of care standards. The APU process is no standardized care process, but the project aims to implement a standardized process to quickly diagnose atraumatic abdominal pain in a symptom-based manner. Due to the very heterogeneous nature of the symptoms of abdominal pain, yet no standardized quality of care outcomes/ key indicators exist. For this reason, we aim to mirror an improvement of care with the three primary outcomes (i.e. acute pain, duration of treatment in the ED, and patient satisfaction). Moreover, we also assess quality of care / patient safety and process quality outcomes (e.g. mortality, course of treatment, ICU stay, process times) in the APU study, as mentioned in the manuscript in Table 1 ‘secondary outcomes’ (see yellow highlighted text, page 8 in revised manuscript).

We hope that we adequately addressed all concerns and we would like to thank the editor and the reviewer once again for the helpful comments and suggestions. We would, however, be happy to consider further specific suggestions you might have.

---

## [Editor Report · Decision Letter 2]

3 Aug 2022

The Abdominal Pain Unit (APU). Study protocol of a standardized and structured care pathway for patients with atraumatic abdominal pain in the emergency department: A stepped wedged cluster randomized controlled trial.

PONE-D-21-38548R2

Dear Dr. Altendorf,

We’re pleased to inform you that your manuscript has been judged scientifically suitable for publication and will be formally accepted for publication once it meets all outstanding technical requirements.

Kind regards,

Steven Eric Wolf, MD

Academic Editor

PLOS ONE
---

## [Editor Report · Acceptance letter]

11 Aug 2022

PONE-D-21-38548R2 

The Abdominal Pain Unit (APU). Study protocol of a standardized and structured care pathway for patients with atraumatic abdominal pain in the emergency department: A stepped wedged cluster randomized controlled trial. 

Dear Dr. Altendorf:

I'm pleased to inform you that your manuscript has been deemed suitable for publication in PLOS ONE. Congratulations! Your manuscript is now with our production department. 

Kind regards, 

on behalf of

Dr. Steven Eric Wolf 

Academic Editor

PLOS ONE